# Stayin’ Alive in Little 5: Application of Sentiment Analysis to Investigate Emotions of Service Industry Workers Responding to Drug Overdoses

**DOI:** 10.3390/ijerph192013103

**Published:** 2022-10-12

**Authors:** Sarah Febres-Cordero, Daniel Jackson Smith

**Affiliations:** 1Nell Hodgson Woodruff School of Nursing, Emory University, Atlanta, GA 30332, USA; 2Fitzpatrick College of Nursing, Villanova University, Villanova, PA 19085, USA

**Keywords:** opioids, service industry workers, sentiment analysis, data-science, mixed-methods

## Abstract

The opioid epidemic has increasingly been recognized as a public health issue and has challenged our current legal, social, and ethical beliefs regarding drug use. The epidemic not only impacts persons who use drugs, but also those around them, including people who do not expect to witness an overdose. For example, in the commercial district of Little 5 Points, Atlanta, GA, many service industry workers have become de facto responders to opioid overdoses when a person experiences an opioid-involved overdose in their place of employment. To provide additional insights into >300 pages of interview data collected from service industry workers that have responded to an opioid overdose while at work, we utilized a mixed-methods approach to conduct this sentiment analysis. First, using R version 4.2.1, a data-science based textual analytic approach was applied to the interview data. Using a corpus algorithm, each line of interview text was characterized as one of the eight following sentiments, anger, anticipation, disgust, fear, joy, sadness, surprise, or trust. Once having identified statements that fit into each of these eight codes, qualitative thematic analysis was conducted. The three most prevalent emotions elucidated from these interviews with service industry workers were trust, anticipation, and joy with 20.4%, 16.2%, and 14.7% across all statements, respectively labeled as each emotion. Thematic analysis revealed three themes in the data: (1) individuals have a part to address in the opioid epidemic, (2) communities have many needs related to the opioid crisis, and (3) structural forces create pathways and barriers to opioid overdose response and rescue. This analysis thematically identified roles service industry workers have in addressing the opioid crisis in Atlanta. Similarly, community needs and barriers to responding to an opioid-involved overdose were characterized. Uniquely, this study found key sentiments related to each of these themes. Future research can leverage these findings to inform the development of overdose prevention and response interventions for service industry works that systematically address common emotions and beliefs trainees may have.

## 1. Introduction

The United States has seen an unprecedented increase in the number of drug overdose deaths, particularly due to the impact of synthetically manufactured opioids [1]. Since 1999, over 800,000 people have died from a drug overdose and, in 2019, 70% of 70,000 overdose deaths involved an opioid [2,3]. In Georgia, the opioid epidemic has been declared a public health emergency with a 207% increase in opioid-related deaths between 2010 and 2020 for a total of 1309 deaths in 2020 [4]. The overwhelming driver of increased deaths due to opioid use is the contamination of the drug supply with synthetic opioids, such as fentanyl, that are 50 times more potent than heroin [5]. One of the most important steps that the State of Georgia has taken to reduce the number of deaths from opioid overdose is the implementation of standing orders, which allow pharmacists and other healthcare professionals to distribute naloxone, an opioid antagonist, to laypersons without the need for a provider’s prescription [6]. The implementation of naloxone standing orders has been shown to increase community availability of this life saving medication [7,8] and decrease opioid-related mortality [9].

Little 5 Points (L5P) is a commercial district located in Atlanta, GA, and is surrounded by many in-town neighborhoods. In response to the opioid epidemic in Georgia, more than 1000 units of naloxone, more commonly known by the brand name Narcan^®^, have been distributed to residents and community members of L5P by the Georgia Department of Public Health and community advocacy organizations, such as the Atlanta Harm Reduction Coalition and Georgia Overdose Prevention [10]. Due to this distribution of naloxone in the L5P community, many businesses now have naloxone available in the workplace and service industry workers are now serving as first responders to overdose when it occurs by a community member in their workplace [11,12]. Restaurant and service industry workers are uniquely positioned to reduce preventable deaths by being prepared to respond to an opioid overdose. For example, in large urban settings as many as 1 in 4 documented overdoses are estimated to occur in public areas like bars, restaurants, and on the street [13,14]. However, overdose response trainings designed for service industry workers require tailored messaging to be impactful, should address emotions or stigma, and meet trainees’ learning needs.

The primary purpose of this study was to analyze the emotions experienced by service industry workers who responded to an opioid overdose at work. This was done by collecting interview data and analyzing it using a blend of data science-based textual analysis techniques with more traditional, qualitative thematic analysis. For this study, we conducted sentiment analysis of emotions from service-industry workers (e.g., restaurant workers/servers, business owners, parking attendants, pharmacy workers, and retail clerks, etc.) who had responded to an overdose in their workplace. We expanded on this identification, by then conducting thematic analysis of exemplar sentences of each of these eight emotions, which provided further insight into the richness of the data.

## 2. Materials and Methods

### 2.1. Parent Study

The parent study employed bricolage, or purposeful mixing of qualitative methods [15], which included elements of ethnography, phenomenology, and grounded theory to describe the experiences of service industry workers who encountered and responded to an opioid overdose in a community setting. Service industry workers were selected due to their close proximity to community opioid-involved overdoses [13,14] and their documented conversion to default first-responders when encountering an overdose in the workplace [11,12]. The parent study utilized the engagement of a community advisory board, participant observation, criterion and snowball sampling, member checks, researcher reflexivity, peer review, triangulation of methods, constant comparison, and two independent coders to ensure interrater agreement of final codes and themes to ensure internal validity [16]. The final sample included 15 service industry workers, working in the retail district of L5P, Atlanta, GA, who participated in interviews with the first author to discuss their experiences of responding to an opioid overdose in their workplace. This sample size was reached due to saturation of the qualitative interviews, which is when no new insights are appearing in the interviews [17,18]. The results from this traditional, qualitative analysis have been presented elsewhere.

Due to the 300 pages of textual data, we desired knowing the additional findings that were present in these interviews. Therefore, we selected a quantitative, data-science based textual analysis which facilitated an efficient and accurate evaluation of the sentiments, or emotions, present in the 300 pages of textual data. As has been applied to other large sets of textual data analyzed via sentiment analysis [19,20], a subsequent qualitative, thematic analysis was then utilized to identify themes within the quantitative sentiment analysis data to provide further insights into the 300 pages of textual data.

### 2.2. Cleaning

When preparing textual data for sentiment analysis, text should be converted to all lower case and any links should be removed [21,22]. Following the methods of Smith et al. [21], stop words and punctuation were retained in the corpus to preserve the meaning of the original text. Stop words are words such as “a” and “the” and provide no context or meaning during frequency analysis. To clean our interview data, a single Microsoft word document containing the interview transcripts was imported into R and, utilizing the *tm* package [23], transformed into a corpus (i.e., a collection of text that is used for textual analysis) and a document term matrix. The conversion to lowercase is essential so that words with different casing (i.e., high and High) are not counted as separate words.

### 2.3. Sentiment Analysis

From the *syuzhet* package [24] and utilizing the specific function ‘get_nrc_sentiment’, which extracts sentiment using the NRC Word-Emotion Association Lexicon, sentiment analysis was performed on the data. The function assigns one of eight emotions (i.e., anger, anticipation, disgust, fear, joy, sadness, surprise, and joy) and one of three overall sentiments (i.e., positive, negative, or neutral) to each word present in the dataset. Words are considered positive when they have a sentiment score >0, neutral when their score equals 0, and negative when their score is <0 [24]. We have chosen to present only the results of the 8 emotions in this paper due to the richness of the analysis.

### 2.4. Thematic Analysis

Following the identification of statements that were labeled as one of eight emotions through the quantitative sentiment analysis process, we conducted thematic analysis using the top ten sentences from each emotion. This portion of the analysis was iterative and utilized inductive reasoning. The eight emotions were utilized as codes, which led to the identification of categories from the sentences identified as each emotion. These categories were then further delineated into themes [16].

## 3. Results

Out of the 330 pages of analyzed interview data, approximately 20.4% of sentences were labeled as trust, 16.2% as anticipation, 14.7% as joy, 13.3% as fear, 10.9% as sadness, 9.6% as anger, 7.5% as surprise, and 7.4% as disgust (Figure 1). A select sample of the sentences rated as each of the 8 emotions are presented in Table 1. Thematic analysis (Figure 2) led to three themes: (1) individuals have a part to play in the opioid epidemic, (2) communities have many needs, and (3) structural forces create pathways and barriers to opioid overdose response and rescue.

## 4. Discussion

### 4.1. Individuals Have a Part to Play in Addressing the Opioid Epidemic

Trust was the most pervasive emotion found in this sentiment analysis. When grappling with the emotions of communities where people are overdosing in a public setting and being rescued by others, we must consider how trust plays out in an overdose rescue attempt. In L5P there is trust in individuals to aid in opioid overdose response and rescue [25]. People who rescue others from an opioid overdose in this setting often rescued as a team with many people involved in a rescue. There is also trust in laws protecting individuals. One such example are Good Samaritan laws that protect individual protection for doing “the right thing” [26]. This was seen in sentences such as “You’re not liable if you do something as a good Samaritan”.

Many participants anticipated who might overdose based on stereotypes used for profiling based on stigma symbols of those who use are thought to use drugs (e.g., abscesses and evidence of injection drug use on arms and legs) [27]. Trust was often conveyed in the notion that service industry workers could potentially identify those who might be in distress by identifying behaviors thought to be predictable by those who use drugs, namely heroin [28]. However, there is some level of profiling that exists when someone walks into an establishment in L5P, which was identified as trusting one’s instincts. If behaviors mimicked those observed during past overdose events, service industry workers often felt they needed to rely on this profiling to protect themselves, their place of employment, and the customers who frequent the area. It is worth noting that this profiling is often confounded by the existence of a homeless population and a population of those with unmet mental health needs, which is now seen as a health disparity issue, particularly regarding race [29].

Although stigma still exists, there are harm reduction efforts underway in L5P that have provided access to free naloxone and syringe exchange services [30]. In L5P over 1000 units of Naloxone have been distributed for free to residents, people who use drugs, and service industry workers with the intention to reduce transmission of blood-borne disease and decrease the number of lives lost from opioid overdose [10,11]. The people who work in L5P acknowledge that with these harm reduction efforts in place it becomes more difficult to stereotype those who use drugs [25]. Nonetheless, these service industry workers have learned to trust their instincts, and to not delay action when an overdose is suspected.

Participants also anticipated that the opioid epidemic has no end in sight and that people who sell drugs are fueling the epidemic, are causing death and despair, and will continue to do so [25]. This was evidenced by sentences such as “You make the choice to sell this, then therefore, you are contributing to the --downfall or eventual death.” However, congruent with the work of Bardwell et al. [31], they often lacked the ability to understand that people who sell drugs on the street level are often people who use drugs themselves and may even trust their drug dealers depending on their relationship with them when purchasing drugs. Conversely, the participants were also keenly aware of the notion that there are roads to recovery if you can destigmatize drug use [32]. It was not uncommon for participants to share their views on the need to decriminalize, and even legalize all drugs to combat stigma surrounding drug use [33].

Joy is also found in the ability of individuals to save lives in the wake of the opioid epidemic, and in the understanding, complex needs of people who use drugs [34,35]. This seen in sentences such as “You’ll be like, OK, that looks like they’re going to treat him for OD.” In our sample, there was a desire to contribute to solutions for those who are part of the community including people who are homeless and those with untreated mental health needs. There is no ignoring the complexities of the social issues facing this commercial district, balancing commercialism and social justice are part of daily life in L5P.

Fears of addiction, death, homelessness, violence, incarceration, overdose, pain, suffering, and people who use and sell drugs are common fears of those on the fringes of our society. Entire books have been dedicated to understanding the complex stigma surrounding these constructs [36]. Fear is a complex human emotion often rooted in a desire to remain safe. In Atlanta, and across the country, the opioid epidemic is beginning to affect all people who use illicit drugs, as the drug supply is tainted with fentanyl, an opioid analog 50 to 100 times more potent than morphine. The diffusion of fentanyl in the drug supply has become painfully obvious as opioid-related overdoses increased by 36% in 2020 partly due to fentanyl contamination of non-opioid drugs [37,38]. Now, not only those who seek out pure opioids are at risk for overdose, but there is an increasing recognition of unintentional polysubstance use’s contributions to the opioid epidemic [39].

### 4.2. Communities Have Many Needs

Service industry workers found joy in their sense of community and a safe place to be different. To understand this emotion, we must recognize that communities are complex. L5P is not only a commercial district located in Atlanta but also serves as a place where people find refuge from the dominant culture [40]. Many subcultures and countercultures exist in L5P. As a primarily independently run commercial district, L5P attracts many who have been expelled from the dominant culture, or who wish to find a way to exist without having to depend on corporate culture. Service industry workers in L5P have many subcultures to choose from when seeking employment there. There is a skateboard culture, a tattoo culture, a co-op culture, a bizarre culture, and an art culture, to name a few. When considering the emotions of the participants it must be noted that people who are drawn to a place like L5P are often considered to be on the fringes of society. In addition to the many cultures that exist in L5P, the area also provides a place of refuge for the homeless, and people with unmet mental health needs. Many would consider L5P a place of liberal views and acceptance [41,42], however, the stigma around drug use and homelessness persists, even in this welcoming harbor.

Joy was also expressed as the ability to have long-term employment (even if service industry workers cannot afford to live close to the workplace), and, in our sample, many of the participants had worked in L5P for 5 to 20 years. This is surprising because L5P is home to many industries, such as restaurants, bars, and retail, where employee turnover is 75% (meaning that 75% of employees in the industry do not stay at their place of employment for over a year) [43]. This is also interesting given that there is joy in long-term employment and job security [44]. However, it can be argued that there is also joy in finding a job where you can be yourself, not be constrained by societal norms, and work in an area that is mostly independently owned. People in L5P can be themselves. In our current society, there are not many places and spaces where those who do not conform are welcome.

### 4.3. Structural Forces Create Pathways and Barriers to Opioid Overdose Response and Rescue

The emotions of anticipation mirror the emotions of trust in many ways. Service industry workers who participated in the parent study anticipated many things including amnesty laws to protect the rescuer, others on the scene who may have drugs on them, and that emergency services would arrive to provide support for the rescue. For example, the anticipation of protection via Good Samaritan laws was seen again with the sentence “You’re not liable if you do something as a good Samaritan.”.

Even though there is no federal law protecting lay persons who respond to opioid overdoses, the State of Georgia has enacted both good Samaritan and medical amnesty laws designed to protect people who rescue someone from an opioid overdose [45]. Before these laws were enacted, people often feared arrest or litigation if they were to become involved in an overdose scenario [46]. Such laws were enacted to increase public trust in calling emergency medical services, which potentially decreases morbidity and mortality related to opioid overdoses [47,48]. Before these laws were enacted, it was common for people to be left to die during an overdose due to fear of arrest [46]. Coupled with these laws, is the trust the community has that when emergency services are called; there is truest that paramedics, firemen, and/or police will arrive on the scene to contribute to the rescue attempt. Given that the national average wait time for emergency medical services is 7 min [49], service industry workers become the default first responders until help arrives. They trust that someone will eventually be there to relieve them from the duty to rescue and that they will not fear litigation if the rescue attempt is a failure.

Naloxone has become the primary means to rescue people from opioid overdoses across the country [50,51]. People who have access to naloxone trust that it will work to revive someone from an overdose. Naloxone was first patented in 1961 to treat constipation from prescription opioids [52] and was being used in emergency departments by the early 1970’s to reverse opioid overdoses [53]. We may now trust in naloxone, but for many decades, people would try to revive those who were overdosing with methods that were effective in anecdotal reports from other people who use drugs, like injecting salt water (which provokes a pain response). Surgeon General Adams was the first to declare the need for all to carry naloxone and as more people are rescued, trust in this drug will surely continue.

Fears surrounding drug use and stigmatized populations are well documented in the literature [28,54,55,56]. Although the participants felt trust and anticipation for police arriving on the scene of an opioid overdose, they also feared them. This is understandable as the relationship between marginalized populations and the police is complex and there is a distrust in law enforcement [57]. This is concerning as previous studies have shown that naloxone administration by police decreases morbidity form overdoses [58]. Participants’ expectations of the police as someone to rescue them from their duty to respond to an overdose is understandable as layperson rescue from an overdose is stressful and can create complex emotions including burden, regret, fear, and anger [35].

Sadness was often expressed in the inability to tackle the overwhelming needs of the community, including the sadness of seeing the homeless population live their entire lives in the community only to die neglected and alone. This was identified with sentences such as “You know, there’s at one point that you come around this way you swing around and come up to get to Freedom Parkway, and there’s always been a homeless population under that bridge area”. Homelessness continues to be a concern in communities across the United States, with estimates that 4.2% of the population will experience homelessness for over one month during their lifetime and 1.5% experiencing homelessness in the past year [59]. Gentrification and cost of living contributed to the feelings of sadness as well. The inability to afford to live in the neighborhood that is part of your identity, was a constant theme in interviews. Gentrification has become a focus of how structural violence contributes to homelessness and is a well-known contributor to homelessness in places like San Francisco, where the cost of living has outpaced the incomes of many within the city [60].

## 5. Conclusions

This mixed-methods analysis of over 300 pages of interviews allowed for further identification of the roles many service industry workers have in addressing the opioid crisis in Atlanta. Uniquely, the quantitative analysis of qualitative data found key sentiments of trust, anticipation, and joy within the interview data. Through additional thematic analysis of these quantitative results, we characterized community needs and barriers to responding to an opioid-involved overdose. Service industry workers in many communities across the country are in close proximity to individuals at risk of experiencing an opioid-involved overdose, either in their workplaces or their larger communities. Future research can leverage these findings to inform the development of overdose prevention and response interventions for service industry workers that systematically address common emotions and beliefs trainees may have.

## Figures and Tables

**Figure 1 ijerph-19-13103-f001:**
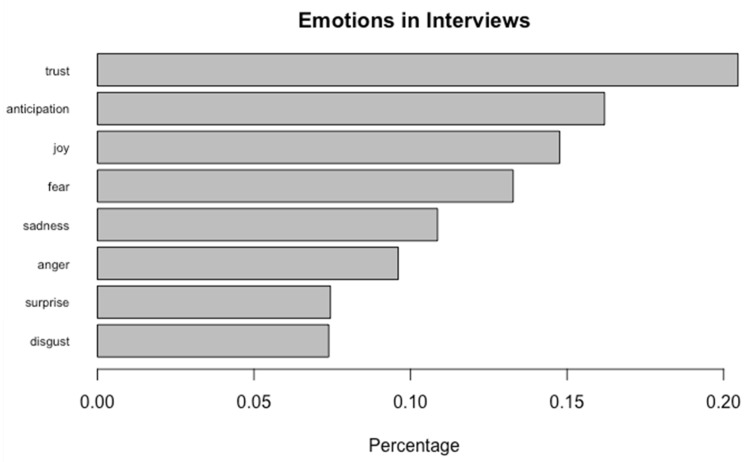
Emotions present in the 300 pages of interview data. 20.4% of sentences were labeled as trust, 16.2% as anticipation, 14.7% as joy, 13.3% as fear, 10.9% as sadness, 9.6% as anger, 7.5% as surprise, and 7.4% as disgust.

**Figure 2 ijerph-19-13103-f002:**
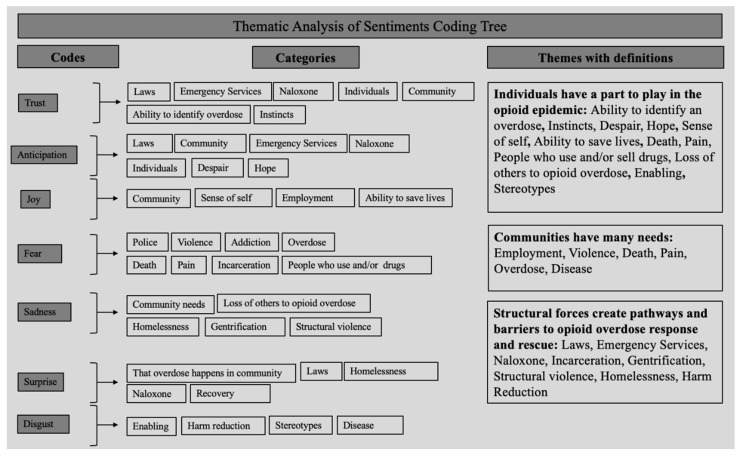
Thematic Analysis of Sentiment Coding Tree; Created by the authors as a part of the iterative thematic analysis process described above. This process resulted in three primary themes: (1) Individuals have a part to play in the opioid epidemic; (2) Communities have many needs; and (3) Structural forces create pathways and barriers to opioid overdose response and rescue.

**Table 1 ijerph-19-13103-t001:** Sentences scored as each of the 8 sentiments.

**Anger Sentences**
“You’d go crazy.” “You wasted it.” “You know, what did you do when you legalized it?” “You know, we’ll just start killing civilians.” “You’ll be like, OK, that looks like they’re going to treat him for OD.” “You’ve got to yell at this guy, hey, stop asking for people’s money, going up to people’s cars.” “You make the choice to sell this, then therefore, you are contributing to the --downfall or eventual death.” “You know, there’s at one point that you come around this way you swing around and come up to get to Freedom Parkway, and there’s always been a homeless population under that bridge area.” “You know, we’ve been fighting this since the early--” “You’re more likely to get shot or hurt in Buckhead at the bar than you are here.”
**Anticipation Sentences**
“You’re not liable if you do something as a good Samaritan.” “You overdosed, and you’re alive but just because like this dude, G, kept him alive.” “You say you can’t have it out in public.” “You know, they have track marks and all that.” “You know, we’ll just start killing civilians.” “You’ll be like, OK, that looks like they’re going to treat him for OD.” “You’re just not having fun.” “You’ve got to yell at this guy, hey, stop asking for people’s money, going up to people’s cars.” “You make the choice to sell this, then therefore, you are contributing to the --downfall or eventual death.” “You want to make them feel like a responsible citizen again, and you do it a little bit at the time.”
**Disgust Sentences**
“You know, people come in all the time and want to buy food for the homeless people.” “You wasted it.” “You know, the cops are hanging around just asking the pharmacist, asking me, asking my co-workers, you know, what happened, yada, yada, yada.” “You know, like, when your skin starts rotting after that?” “You know, shit like that.” “You’ll be like, OK, that looks like they’re going to treat him for OD.” “You make the choice to sell this, then therefore, you are contributing to the --downfall or eventual death.” “You know, there’s at one point that you come around this way you swing around and come up to get to Freedom Parkway, and there’s always been a homeless population under that bridge area.” “You know, shipping containers that are no longer in use can be converted to living spaces to help reduce the homeless.” “You know, it’s like the spider web.”
**Fear Sentences**
“You’d go crazy.” “You know, the cops are hanging around just asking the pharmacist, asking me, asking my co-workers, you know, what happened, yada, yada, yada.” “You know, what did you do when you legalized it?” “You know, we’ll just start killing civilians.” “You’ll be like, OK, that looks like they’re going to treat him for OD.” “You’re just copping, and you’re just cop, gotta cop, gotta get some more, buy a lot.” “You’ve got to yell at this guy, hey, stop asking for people’s money, going up to people’s cars.” “You make the choice to sell this, then therefore, you are contributing to the --downfall or eventual death.” “You know, there’s at one point that you come around this way you swing around and come up to get to Freedom Parkway, and there’s always been a homeless population under that bridge area.” “You’re more likely to get shot or hurt in Buckhead at the bar than you are here.”
**Joy Sentences**
“You probably have to have two jobs just to afford to live around here, depending on what kind of job you have.” “You’re not liable if you do something as a good Samaritan.” “You overdosed, and you’re alive but just because like this dude, G, kept him alive.” “You know, what did you do when you legalized it?” “You’ll be like, OK, that looks like they’re going to treat him for OD.” “You’re just not having fun.” “You’ve got to yell at this guy, hey, stop asking for people’s money, going up to people’s cars.” “You need to have a little welcoming spot for your customers to come in, park, and feel safe.” “You know, there’s at one point that you come around this way you swing around and come up to get to Freedom Parkway, and there’s always been a homeless population under that bridge area.” “You know, we need facilities where if there’s-- we need mental health, safe use.”
**Sadness Sentences**
“You’d go crazy.” “You know, that generation, the older-- ?” “You know, the cops are hanging around just asking the pharmacist, asking me, asking my co-workers, you know, what happened, yada, yada, yada.” “You know, we’ll just start killing civilians.” “You know, you’d also get tax revenue on it, which could probably help wipe out the debt, I mean, if we’re not kidding.” “You’ll be like, OK, that looks like they’re going to treat him for OD.” “You make the choice to sell this, then therefore, you are contributing to the --downfall or eventual death.” “You remember R, with the big plats in his hair and then they fell off and stuff?” “You know, there’s at one point that you come around this way you swing around and come up to get to Freedom Parkway, and there’s always been a homeless population under that bridge area.” “You’re more likely to get shot or hurt in Buckhead at the bar than you are here.”
**Surprise Sentences**
“You just wanted to rubber neck the accident.” “You’re not liable if you do something as a good Samaritan.” “You’ll be like, OK, that looks like they’re going to treat him for OD.” “You’ve got to yell at this guy, hey, stop asking for people’s money, going up to people’s cars.” “You know, I’ve tried to guess, and I’m saying maybe 40.” “You make the choice to sell this, then therefore, you are contributing to the --downfall or eventual death.” “You’re more likely to get shot or hurt in Buckhead at the bar than you are here.” “You know, I haven’t gotten to the point where my emails say, my pronouns are, because I had to deal with certain types of people that don’t understand that.” “You know, here’s some money.” “You have to-- I mean, you kind of do have to have someone help you, or you have to have, I guess, methadone or something to, like, help wean you off of it.”
**Trust Sentences**
“You probably have to have two jobs just to afford to live around here, depending on what kind of job you have.” “You’re not liable if you do something as a good Samaritan.” “You overdosed, and you’re alive but just because like this dude, G, kept him alive.” “You think drugs should be legal.” “younger than her, she was real cooky.” “You’ll be like, OK, that looks like they’re going to treat him for OD.” “You’re just copping, and you’re just cop, gotta cop, gotta get some more, buy a lot.” “Your whole bowel, everything’s just fucked in your system.” “You’ve got to yell at this guy, hey, stop asking for people’s money, going up to people’s cars.” “You want to make them feel like a responsible citizen again, and you do it a little bit at the time.”

## Data Availability

The data presented in this study are available on reasonable request from the corresponding author. The data are not publicly available due to privacy concerns.

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
