# Peer review of "Stayin’ Alive in Little 5: Application of Sentiment Analysis to Investigate Emotions of Service Industry Workers Responding to Drug Overdoses"

_ijerph, 2022, doi:10.3390/ijerph192013103_

Round 1
Reviewer 1 Report
There is no doubt that the title captures and draws public attention to this issue and that the study discusses very important workplace issues workers have had to deal with.
1- This manuscript needs a graphical abstract to explain its contents.
2- Line 38-39, "In Georgia, the opioid epidemic has been de-38 clared a public health emergency. Between 2010 and 2020 there was a 207% increase in 39 opioid-related deaths [4]". a- How many cases were there and how many are there now? b- Why is there an increase in cases? It would be useful if you provided some details in this section of your paper.
3- Line 40-44, What are the other acts? You could briefly mention it and explain why "allowing pharmacists and other healthcare professionals to distribute naloxone" is important.
4- Line 53-54, "Service industry workers"? It is not clear. "Service industry workers" should be described with some examples.
5- Line 60-67, Why do you choose this group? Are they obligated by law to respond to the over-dose case at their work area? Your study is focused on this group; It is better to enrich the reader with some background on this issue.
6- Line 77, "The final sample included 15 service industry workers" is it stylistically enough? If Yes explain.
7- Figure 1 and 2 should be described in detail in the legend of the picture.
8- In the discussion part, I am concerned that the authors discussed major issues without incorporating their results. A discussion of how the results relate to the problem was not included in any of the sections. This point must be clarified by the author and they should discuss their results in a way that makes sense.
9- Again, the conclusion part was unclear to me. It would be helpful if the main findings were mentioned.
Author Response
Please, see the attached file.

Reviewer 2 Report
I think that the motivation is really relevant.
The methodology was really appropriate using the textual analysis. After separating the sentences, it was really worthy to categorize the sentiments.
My only doubt is if the Figure 2 was developed by the authors of these papers or if it was obtained in the literature. I ask this because it seems a photo of other book. I think you developed, so it would be better if you say in the title or footnote "developed by the authors" or some like this.
As statistician , I learned and liked the methodology and that you included all packages used. Please, cite the packages as
@Manual{,
title = {Syuzhet: Extract Sentiment and Plot Arcs from Text},
author = {Matthew L. Jockers},
year = {2015},
url = {https://github.com/mjockers/syuzhet},
}
This is a recognition for the authors that developed and let this package available for free use in R.
Round 2
Reviewer 1 Report
Well done